# Potential Molecular Mechanisms behind the Ultra-High Dose Rate “FLASH” Effect

**DOI:** 10.3390/ijms232012109

**Published:** 2022-10-11

**Authors:** Eva Bogaerts, Ellina Macaeva, Sofie Isebaert, Karin Haustermans

**Affiliations:** 1Department of Oncology, KU Leuven, 3000 Leuven, Belgium; 2Institut de Recherche Expérimentale et Clinique (IREC), Université Catholique de Louvain, 1200 Woluwé-Saint-Lambert, Belgium; 3Department of Radiation Oncology, University Hospitals Leuven, 3000 Leuven, Belgium

**Keywords:** FLASH radiotherapy, ultra-high dose rate, radiobiology, normal tissue sparing, oxygen, immune system, mitochondria

## Abstract

FLASH radiotherapy, or the delivery of a dose at an ultra-high dose rate (>40 Gy/s), has recently emerged as a promising tool to enhance the therapeutic index in cancer treatment. The remarkable sparing of normal tissues and equivalent tumor control by FLASH irradiation compared to conventional dose rate irradiation—the FLASH effect—has already been demonstrated in several preclinical models and even in a first patient with T-cell cutaneous lymphoma. However, the biological mechanisms responsible for the differential effect produced by FLASH irradiation in normal and cancer cells remain to be elucidated. This is of great importance because a good understanding of the underlying radiobiological mechanisms and characterization of the specific beam parameters is required for a successful clinical translation of FLASH radiotherapy. In this review, we summarize the FLASH investigations performed so far and critically evaluate the current hypotheses explaining the FLASH effect, including oxygen depletion, the production of reactive oxygen species, and an altered immune response. We also propose a new theory that assumes an important role of mitochondria in mediating the normal tissue and tumor response to FLASH dose rates.

## 1. Introduction

Radiotherapy (RT) contributes to the treatment of approximately half of all cancer patients, thus playing a major role in cancer management along with chemotherapy and surgery [1,2,3,4,5,6,7,8]. The ultimate goal of RT is to deliver lethal doses of ionizing radiation to a tumor while minimizing the toxicity to normal tissues and critical organs. Over the past decades, important technological advances in RT such as image-guided RT, intensity-modulated RT, stereotactic body RT, and particle therapy have greatly improved the accuracy of RT and hence the therapeutic index [2,9,10,11]. Despite this progress, the RT dose necessary to reach tumor control is still limited by the radiation-induced toxicities to the surrounding healthy tissues [12,13]. Moreover, a significant proportion of cancers remain intrinsically radioresistant, requiring escalation of the dose to reach tumor control [14]. Therefore, more effective and better-tolerated radiotherapeutic strategies need to be developed that spare the surrounding healthy tissues while maintaining or even improving the anti-tumor effect [12,15]. 

In recent years, a novel treatment approach called FLASH RT has attracted great attention due to its ability to remarkably protect healthy tissues from radiation-induced damage while preserving the same tumor control when compared to conventional RT (CONV RT) in a preclinical setting [15,16,17,18]. This phenomenon is known as the FLASH effect and was rediscovered in 2014 by the group of Favaudon and Vozenin [17]. FLASH RT involves the delivery of radiation doses at ultra-high dose rates (≥40 Gy/s)—several thousand times higher than what is currently used in clinical practice for CONV RT (≈5 Gy/min) [12,19,20]. Several research groups have already been able to confirm the FLASH effect in various in vivo models including zebrafish [21,22], mice [16,17,21,23,24,25,26,27,28,29,30,31,32,33,34], cats, and minipigs [15]. In 2019, a first patient with T-cell cutaneous lymphoma was successfully treated with FLASH RT [35]. Even though the surprising normal-tissue-sparing effects are most thoroughly characterized for electron irradiations, proton and X-ray ultra-high dose rate irradiation also has been shown to result in less normal tissue toxicity and similar tumor control compared to CONV RT [16,36,37,38,39]. The in vitro and in vivo experimental studies on FLASH RT are listed in Table 1 and Table 2, respectively.

Although FLASH RT could become one of the most significant innovations in the radiation oncology field, the exact biological mechanisms underlying its unique effects remain to be elucidated. The current hypothesis gaining the most ground suggests that ultra-high dose rate irradiation induces an acute oxygen depletion, causing a transient radioprotective hypoxia in irradiated normal tissues [12]. However, it is not clear why in vivo tumors do not benefit from the FLASH effect. An altered immunological response following FLASH RT has also been suggested as a potential mechanism that explains the increased therapeutic ratio of FLASH RT compared to CONV RT [50,51,52,53]. The short exposure time, an important feature of FLASH RT, would significantly reduce the proportion of circulating immune cells being irradiated and killed, leading to a more functional immune system that can more effectively repair radiation-induced damage to normal tissue [50]. In addition, the differential immunological response to FLASH RT might contribute to the decrease in inflammation observed in healthy tissues and the induction of a more potent anti-tumor immunity, resulting in greater tumor control.

In this review, we provide more detail about various potential mechanisms for the FLASH effect, including the ones already mentioned on oxygen and a modified immune response. Furthermore, we also highlight a more recent theory about a differential effect on mitochondria following FLASH versus CONV dose rate irradiation.

## 2. Mechanisms for the FLASH Effect

### 2.1. The Role of Oxygen in the FLASH Effect

#### 2.1.1. Impact of Oxygen Concentration

To date, the exact biological mechanisms responsible for the FLASH effect are still not completely understood, but the most prominent hypothesis suggests that the normal-tissue-sparing effect may be attributed to a prompt radiochemical depletion of oxygen [51,60,61,62]. In general, indirect DNA damage, which is the most common type of DNA damage after low linear energy transfer (LET) irradiation, occurs due to water radiolysis and subsequent generation of free radicals [63,64]. When interacting with DNA, free radicals cause damage that can be easily reversed by antioxidants. However, the presence of dissolved molecular oxygen in the cell results in fixation of the DNA damage, making it more difficult to repair. Thus, in the absence of oxygen, cells are less susceptible to radiation-induced lethal DNA damage. This is why hypoxic tumors are often highly radioresistant. After exposure to ionizing radiation, ionization reactions cause small depletions in the available oxygen. The oxygen depletion theory states that more oxygen is consumed at ultra-high dose rates because considerably more electrons are liberated per unit of time, resulting in substantially more ionization than that produced at CONV dose rates [20]. Moreover, during the very short exposure time frame of FLASH RT, the depletion in local oxygen might occur faster than any tissue reoxygenation kinetics, resulting in a transient state of radiobiological hypoxia and making the normal tissues more radioresistant [65].

A relationship between increasing dose rates and oxygen depletion was first suggested by Dewey and Boag, who demonstrated that the survival curves of bacteria irradiated at ultra-high dose rates were comparable to those of bacteria irradiated in hypoxic conditions [66]. They believed that the first few kilorads of a FLASH electron pulse might remove dissolved oxygen in bacteria via radiation-induced reactions. This would generate a nearly anaerobic environment in which the bacteria would receive the remainder of the irradiation dose. As the duration of the pulse was only 2 µs, dissolved extracellular oxygen was not able to penetrate the bacterial cell by diffusion. In the following years, the oxygen depletion theory was further investigated using bacteria [67,68], mammalian cell lines [69,70], and small animals [71]. Recently, Adrian and colleagues confirmed the role of oxygen concentration in the differential response of prostate cancer cells to FLASH or CONV RT treatment [54]. Cells exposed to electron FLASH dose rates (600 Gy/s) showed significantly higher survival versus CONV (0.23 Gy/min) irradiated cells at physiologically relevant oxygen concentrations (1.6%, 2.7%, and 4.4% O_2_), but not at higher oxygen levels (8.3% and 20% O_2_) [54]. They attributed the obtained results to the transient lowering of the partial oxygen pressure due to high-dose-rate irradiation, thereby increasing the cells’ radioresistance. Similar results were found by Khan et al., who observed a threefold higher survival for FLASH (90 Gy/s)-irradiated multicellular tumor spheroids characterized by a hypoxic core compared to CONV (0.075 Gy/s)-treated spheroids, whereas 2D cultured cells irradiated under normoxic conditions did not show significant differences in survival after FLASH vs. CONV irradiation [56].

While there is no doubt that oxygen plays a role in mediating the FLASH effect, a complete depletion of oxygen seems unlikely to occur after ultra-high dose rate irradiation. Radiolysis-based models built by Boscolo et al. [72] and Labarbe et al. [73] assumed that oxygen depletion was too slow and insufficient to fully account for the FLASH effect and did not support radiation-induced transient hypoxia as the dominant mechanism. The models showed that the in vitro results obtained by Adrian et al. could not be solely explained by a radiolytic consumption of oxygen because a residual amount of oxygen remained present.

Nevertheless, there might be hypoxic stem cell niches in well-oxygenated normal tissues that can be spared following FLASH irradiation via oxygen depletion, as proposed by Pratx and colleagues [74,75]. Although these regions are much smaller than the normoxic regions, the fraction of cells that survive ionizing radiation depends on the dose and oxygen levels, so the radiotherapy response of tissue will therefore be determined by the most hypoxic cells.

An important question remained regarding whether tumor stem cells residing in hypoxic niches would also be spared by ultra-high dose rate irradiation [76]. By modeling the impact of spatial oxygen heterogeneity on radiolytic oxygen depletion, Taylor et al. demonstrated that the relative increase in cell survival after FLASH vs. CONV RT was more pronounced for better-oxygenated normal tissues than for hypoxic tumors [77]. They attributed this differential response to the fact that FLASH RT lowered the mean tissue oxygen partial pressures (oxygen depletion) by an amount dependent upon the initial mean oxygen partial pressure. In other words, tissues with a low mean partial pressure (hypoxic tumors) might be less susceptible to FLASH because the small amount of oxygen depletion would only result in a limited sparing of hypoxic stem cell niches. The importance of the initial tissue oxygenation level on the FLASH effect was also recently demonstrated by a mathematical damage model using oxygen-dose histograms [78]. The authors estimated the relative sparing from DNA damage at two different initial oxygenation levels: 20 mmHg pO_2_ (representing well-oxygenated tissues) and 2 mmHg pO_2_ (representing hypoxic tumor tissues). Depending on the FLASH dose delivered, the sparing occurred to a larger extent in one tissue over the other. At low radiation doses, the FLASH effect was greater for the tissues with lower oxygenation levels, whereas above 10 Gy, the well-oxygenated tissues exhibited a larger amount of sparing, indicating that a threshold dose for the FLASH effect might exist. However, as most in vivo data do not show any FLASH effect in tumors at all, there must be additional factors besides tissue oxygenation that are responsible for the observed differences between normal and cancerous tissues following FLASH RT.

#### 2.1.2. Differences in Redox Metabolism between Normal and Tumor Tissues

An interesting theory based on intrinsic differences in redox metabolism was offered by Spitz et al. that might explain the preserved tumor control observed after FLASH RT [79]. Since normal cells have lower pro-oxidant burdens during normal steady-state metabolism and a greater ability to enzymatically reduce peroxidized compounds compared to cancer cells, this theory assumes that FLASH-induced organic hydroperoxides are more efficiently removed by normal vs. tumor tissues. Moreover, normal cells have reduced levels of redox-active labile iron, which catalyzes the production of hydroxyl radicals via Fenton-type chain reactions, and thus are able to more easily regulate and sequester labile iron pools. Thus, if ultra-high dose rate irradiation could deplete all of the local tissue oxygen by converting oxygen in both normal and cancer tissues to hydroperoxides, then the faster elimination of these toxic hydroperoxides via antioxidant pathways in normal cells and the excess of labile iron in cancer cells could explain the sparing of normal cells and killing of cancer cells after FLASH RT. In contrast, CONV-dose-rate irradiation generates much lower total yields of free radicals and hydroperoxides (four orders of magnitude), resulting in a negligible differential effect in redox metabolism between normal and cancer tissue.

#### 2.1.3. Radical–Radical Interactions

Other reasons why the transient radioprotective hypoxia hypothesis has been seriously questioned lately could be the fact that this theory may not account for the results of in vitro studies that used normal cells maintained under atmospheric O_2_ levels. Several FLASH in vitro studies have reported significant sparing of normal cells from induction of chromosomal aberrations and limited radiation-induced senescence [31,57,80,81,82,83,84]. Acharya and colleagues investigated the effects of single vs. multiple pulses of 7 MeV electrons on micronuclei induction in in vitro irradiated human blood lymphocytes over a wide range of dose rates per pulse (10^6^–3.2 × 10^8^ Gy/s) [83]. A significant decrease (37–72%) in the frequency of micronuclei was found with an increasing dose rate when the dose was delivered by a single pulse; the higher the dose (2–8 Gy), the higher the observed effect. However, the micronuclei yield was not reduced when a dose above 4 Gy was delivered in multiple nanosecond pulses. The authors suggested increased probability of radical–radical recombination and thus less radical–DNA target interactions as a possible explanation for the observed reduction in micronuclei yield after single pulses of ultra-high dose rate radiation.

It has been reported that in pulse radiolysis, a competition exists between self-recombination of carbon-centered radicals (R•) and the reaction of R• with oxygen [73]. The rates of the reaction of R• with oxygen are proportional to the radical concentration but the rate of radical–radical recombination is proportional to the square of the radical concentration [73]. As mentioned by Wardman, short pulses of ultra-high dose rate irradiation produce a higher transient free radical concentration within a shorter time window than for CONV RT treatment [85]. Thus, as the transient concentration of radical species increases following FLASH RT and the rate of radical recombination increases proportionally more than the rate of reaction with oxygen, less R• will be available to interact with oxygen and DNA, leading to less oxidative tissue damage. This hypothesis has been strengthened by a recent study using Monte Carlo simulations of ionizing radiation track structures in water in which higher radical recombination rates were found at FLASH dose rates (40 Gy/s) [86].

A more recent study on normal human lung fibroblasts and human lung adenocarcinoma cells cultured under normoxic conditions demonstrated significantly less 53BP1 foci formation in the normal cells exposed to an electron dose of 5 Gy at ultra-high dose rates (106 Gy/s) compared to CONV-treated normal cells [31]. This difference was not observed in the cancer cells grown under similar culturing conditions. Furthermore, FLASH RT could spare primary human pulmonary basal stem cells from radiation-induced differentiation and cell death as opposed to CONV RT. Specifically, 53BP1 is a DNA double-strand break (DSB) repair protein with a crucial role in both homologous recombination and non-homologous end-joining and is considered to promote microhomology end-joining (MMEJ) in G1 phase cells [87]. Based on these results, Labarbe et al. suggested that the differential effect between normal and cancer cells to FLASH RT might be explained by a combination of two processes; i.e., by R• recombination, causing fewer DNA DSBs that are detected by 53BP1, and by a repair defect specific to cancer cells in the G1 phase [73]. Importantly, MMEJ promotion by 53BP1 in G1 was only seen in cells where *BRCA1* was normally expressed [87]. As tumor cells often harbor *BRCA1* mutations, DSBs will not be properly repaired, inducing more cell death and genomic instability than in normal cells.

However, one study also demonstrated the sparing FLASH effect in cancer cells at low doses under normoxic in vitro conditions [55]. Six cancer cell lines (MDA-MB-231, MCF7, WiDr, LU-HNSCC4, and HeLa (early passage and subclone)) showed a tendency toward increased survival after FLASH (≥800 Gy/s) vs. CONV (14 Gy/min) dose rates, with significant differences for four cell lines. Cell cycle distributions and 53BP1-foci formation were similar between the two treatment modes.

These inconsistent in vitro results could indicate differences in intrinsic biological susceptibility to ultra-high dose rate irradiation. This assumption was also made by a recent in vivo study in which only two out of three patient-derived xenograft mouse models of T-cell acute lymphoblastic leukemia (T-ALL) were more responsive to FLASH (200 Gy/s) total body irradiation (TBI) than CONV (<0.072 Gy/s) TBI, whereas the third case of T-ALL was more sensitive to CONV TBI [26]. This observation suggested that individual intrinsic factors can differentially affect the sensitivity of human T-ALL to FLASH and CONV RT. Another reason for the conflicting in vitro and in vivo outcomes was the use of different physical parameters by different research groups. The key beam parameters required to drive normal tissue protection such as total dose, dose rate within the pulse, beam-on time, and pulse frequency varied significantly in the different FLASH studies and were sometimes not properly documented, which may have complicated the interpretation and comparison of the results.

### 2.2. The Role of the Immune System in the FLASH Effect

In addition to the role of oxygen and radical–radical interactions in mediating the FLASH effect, it is likely that a systemic effect that includes inflammatory reactions and a differential immune response determines both normal tissue and tumor responses to FLASH RT in the complex in vivo environment.

#### 2.2.1. Sparing of T Lymphocytes

Ionizing radiation causes the initiation and modulation of inflammatory/immune responses in the irradiated tissues, resulting in normal tissue complications such as fibrosis [88]. However, when the radiation dose is delivered at ultra-high dose rates, less inflammation and fewer fibrotic lesions were observed in irradiated animals, implicating a differential immunological response after FLASH vs. CONV RT [16,17,38,89]. This altered immunological reaction could partially be explained by the shorter exposure time of FLASH RT (<1 s) that would significantly reduce the fraction of circulating lymphocytes being irradiated and killed, thereby sparing the immune system more than CONV RT [50,51,52,53]. This hypothesis has been strengthened by a recent computational study in which the effect of the RT dose rate on immune-cell killing was modeled [50]. The authors calculated an impressive decrease in the percentage of circulating immune cells killed in the irradiated volume from 95% at CONV dose rates (<5 Gy/min) to only 5–10% at ultra-high dose rates (>200 Gy/min) [50]. A dose rate threshold of 40 Gy/s was determined for mice, which was consistent with FLASH dose rates reported in preclinical studies and was about one order of magnitude lower for humans than for mice. The lymphocyte-sparing effect increased in a dose-dependent manner (5–50 Gy/fraction) and almost completely disappeared at 2 Gy/fraction [50]. This increased sparing of immune cells by FLASH RT may lead to a better functioning immune system that can more effectively repair RT-induced normal tissue damage [90]. Additionally, the shorter exposure time and, as a result, the smaller irradiated volume of the blood pool could induce fewer chromosomal aberrations in peripheral blood lymphocytes, resulting in a decreased risk of developing secondary malignancies [51,52,91].

The FLASH-mediated sparing of the highly radiosensitive lymphocytes may not only have a positive impact on normal tissue protection, but also on tumor control. Lymphocytes are essential players in anti-tumor immunity; maintaining a pool of functioning lymphocytes in the circulation might contribute to better treatment outcomes [92]. A great number of clinical studies have reported a clear association between severe radiation-induced lymphopenia (reduced lymphocyte levels) and tumor progression and reduced survival in patients with solid tumors [53,93,94,95]. Moreover, emerging evidence suggests that higher levels of tumor-infiltrating lymphocytes are positively correlated with a favorable prognosis [96]. Excitingly, Rama et al. found an increased recruitment of CD3^+^ T lymphocytes into the tumor microenvironment of FLASH-irradiated mice compared to CONV-treated animals [37]. In this study, a syngeneic, orthotopic C57B1/6J mouse model of non-small cell lung cancer was used to compare the efficacy of 18 Gy single-dose proton irradiation delivered at FLASH (40 Gy/s) versus CONV dose rates on tumor eradication. Interestingly, proton-FLASH-treated mice had significantly smaller lung tumors than CONV-treated animals. Moreover, immunostaining on harvested tumor sections showed higher CD3^+^ T cell levels, including both CD4^+^ and CD8^+^ cells, in the tumors treated with FLASH RT compared to CONV RT. This increased lymphocyte infiltration into the tumor core was also observed by Kim and colleagues, who found elevated levels of cytotoxic CD8α^+^ T cells in mouse lung tumors after FLASH (352.1 Gy/s) vs. CONV RT (0.06 Gy/s) [40]. Together, these data suggested that improved lung tumor control by FLASH radiation might be the result of increased recruitment of T lymphocytes into the tumor.

#### 2.2.2. Less TGF-β Production

Another explanation for the increased normal tissue protection by FLASH RT might be the altered expression of certain cytokines following ultra-high dose rate irradiation. Transforming growth factor beta (TGF-β), a multifunctional cytokine with potent pro-fibrotic and pro-oxidant activity, is typically elevated in response to radiation-induced DNA damage, thereby playing a critical role in normal tissue injury [97,98]. Reduced TGF-β activation and fibrosis were observed in multiple studies involving FLASH RT. An in vitro study by Buonanno et al. demonstrated a significant reduction in the expression of TGF-β1 in human lung fibroblasts exposed to a proton dose of 20 Gy delivered at FLASH dose rate (1000 Gy/s) compared to a CONV dose rate (0.2 Gy/s) [57]. A ~6.5-fold increase in TGF-β1 expression was observed in the CONV-treated cells compared to the sham-irradiated cells while only a ~1.8-fold induction was found in the FLASH-irradiated cells at 24 h post-irradiation, indicating that FLASH RT might prevent the induction of markers that mediate radiation-induced inflammation. Reduced TGF-β activity was also previously observed in FLASH-irradiated mice that underwent whole thorax irradiation with a single fraction of 17 Gy [17]. FLASH RT (≥40 Gy/s) could protect mice from pulmonary fibrosis and prevent activation of the TGF-β/SMAD signaling cascade in blood vessels and bronchi compared to CONV RT (≤0.03 Gy/s). Similar results were obtained by Cunningham et al., who found significant decreases in both plasma and skin levels of TGF-β1 and skin toxicity in mice following FLASH (≥57 Gy/s) vs. CONV (1 Gy/s) proton irradiation [89].

While TGF-β acts as a tumor suppressor in normal and early-stage cancer cells by promoting apoptosis and inhibiting cell cycle progression, it mainly serves as a tumor promotor in late-stage cancer cells by inducing proliferation, invasion, angiogenesis, metastasis, and immune suppression [99]. This altered TGF-β pathway allows malignant cells to escape from TGF-β’s growth-suppressive effects, giving them a selective growth advantage and making them more radioresistant [100,101]. TGF-β also stimulates the phosphorylation of DNA repair enzymes that are crucial for initiating the DNA damage response in cancer cells following exposure to ionizing radiation [102]. If FLASH dose rates have the potential to reduce TGF-β induction not only in normal tissues compared to CONV dose rates but also within the tumor microenvironment (TME), cancer cells would become more radiosensitive and tumor control could be improved (Figure 1). Moreover, a reduction in TGF-β levels after FLASH RT might result in a decreased generation of immunosuppressive regulatory T cells (Tregs). TGF-β facilitates the differentiation of Tregs, which infiltrate into the TME and downregulate the anti-tumor immune response of cytotoxic CD8^+^ T cells [101,103]. Thus, FLASH radiation could have the ability to prevent TGF-β secretion and recruitment of Tregs into the tumor core, thereby increasing anti-tumor immunity compared to CONV RT. However, further research is necessary to confirm the role of TGF-β in the regulation of the reported FLASH effects.

Furthermore, it has been demonstrated that FLASH irradiation can significantly reduce the induction of other cytokines involved in radiation-induced inflammatory processes compared to CONV-dose-rate irradiation [32]. A preclinical study by Simmons et al. that evaluated the impact of whole-brain FLASH RT on hippocampal dendritic spines and neuroinflammation found a significant increase in 5 out of 10 studied pro-inflammatory cytokines (interleukin (IL)-1β, IL-4, IL-6, tumor necrosis factor (TNF)α, and KC/GRO) tested in the hippocampus of C57BL6/J mice after CONV RT (0.13 Gy/s). In contrast, FLASH dose rates (200–300 Gy/s) caused a significant increase in only three cytokines (IL-1β, TNFα, and KC/GRO) and to a lesser extent than CONV dose rates. This trend in reduced levels of inflammation-related markers following FLASH was also observed in the aforementioned study by Cunningham and colleagues. They detected a significant decrease in the C-X-C motif chemokine ligand 1 (CXCL1) and granulate-colony stimulating factor (G-CSF) levels in the blood of FLASH vs. CONV-irradiated animals, while granulate-macrophage-CSF (GM-CSF) levels were higher after FLASH RT [89]. G-CSF and GM-CSF are required for granulocyte production and differentiation whereas CXCL1 serves as a neutrophil attractant [104,105]. Previously, a negative correlation was also found between the GM-CSF/G-CSF ratio measured in the serum of patients with cystic fibrosis and the degree of tissue toxicity [106]. These findings indicated that the differential effects of FLASH vs. CONV RT on normal tissues occur at the level of induction of inflammatory cytokine responses.

#### 2.2.3. Reduced Microglia Activation

The reduced inflammation observed in normal tissues following FLASH vs. CONV RT is most likely mediated by many different immune factors, including macrophages. Upon exposure to ionizing radiation, M1 macrophages are activated to secrete pro-inflammatory cytokines that induce inflammation and an initial anti-tumor immune response [107]. In a later stage, the more radioresistant M2 macrophages are recruited to the sites of injury to secrete profibrotic cytokines, including TGF-β, to promote fibrogenesis [108]. M2 macrophages are also responsible for the generation of an immunosuppressive pro-tumoral microenvironment. Therefore, targeting macrophages to limit or treat radiation-induced tissue toxicity has become a promising strategy to increase the therapeutic index.

Although the impact of FLASH RT on peripheral macrophages still needs to be elucidated, a differential effect on microglia, a specialized population of macrophage-like cells in the central nervous system, between FLASH RT and CONV RT has already been confirmed in multiple in vivo studies [21,30,32,109]. Following cranial irradiation, microglia are activated to release pro-inflammatory cytokines such as IL-1β, IL-6, and TNFα in the hippocampus and other brain regions, eventually resulting in neuroinflammation [32]. Simmons et al. observed significantly higher levels of the microglial protein CD68, a biomarker of microglial activation, in CONV-treated mice (379 CD68^+^ cells/mouse) compared to the control group (132.2 CD68^+^ cells/mouse) at 10 weeks post-irradiation [32]. However, whole-brain FLASH RT could reduce CD68 expression (255.4 CD68^+^ cells/mouse) in comparison to CONV RT, ultimately leading to decreased neuroinflammation and neurocognitive deficits. Recently, a similar neuroprotective effect was shown after ultra-high dose rate irradiation of the radiosensitive juvenile mouse brain [30]. CD68^+^ microglial cells were significantly increased in CONV (0.1 Gy/s)-treated animals, whereas the number of CD68^+^ cells in FLASH (4.4 × 10^6^ Gy/s)-irradiated mice was statistically similar to the control group. A possible explanation for the reduced microglia activation after FLASH could be a decrease in RT-induced ROS, which limited microglia transition to a chronically activated state [21]. These studies demonstrated that FLASH brain irradiation might be effective in sparing hippocampus-dependent spatial learning and memory [32].

In addition to their inflammatory role, macrophages are also involved in tumor development and progression by encouraging de novo angiogenesis, metastasis, and remodeling of the stromal matrix [110]. These tumor-associated macrophages (TAMs) are recruited from tissue-resident macrophages or circulating bone-marrow-derived monocytes (BMDMs) to the tumor site [111,112]. A recent study investigating local differences in TAM recruitment and evolution in glioblastoma showed that TAMs that originated from BMDMs dominated the tumor core and exhibited an inflammatory M1-like phenotype while resident microglia-derived TAMs were more abundant in the tumor periphery and evolved toward an immunosuppressive M2 type [113]. As microglia activation was found to be significantly reduced in healthy mice after FLASH RT compared to CONV RT, it can be assumed that this reduction also causes a decrease in the number of microglia-derived TAMs in tumor-bearing mice. In this way, FLASH RT could limit TAM-mediated immunosuppression and tumor regrowth seen following CONV irradiation. However, further investigations are essential to validate this hypothesis.

### 2.3. The Potential Role of Mitochondria in the FLASH Effect

An interesting hypothesis that is yet to be further explored is that mitochondria-mediated inflammation and apoptosis might be reduced after FLASH RT.

In addition to their essential role as powerhouses of the cell, mitochondria are also highly involved in redox metabolism, Ca^2+^ homeostasis, cell signaling, and apoptosis [114]. Moreover, these organelles have recently been identified as major regulators of the innate immune system and inflammatory processes [115]. Mitochondria are the primary source of intracellular ROS (mtROS), which serve as signaling molecules for cell growth and survival under physiological conditions but cause oxidative damage to proteins, mitochondrial DNA (mtDNA), and lipids under cellular stress [116]. When a cell is exposed to ionizing radiation, the excessive production of mtROS can result in severe and irreversible damage to the mitochondria, which leads to mitochondrial outer membrane permeabilization (MOMP), a process induced by the pro-apoptotic Bcl-2 proteins BAX and BAK [117,118]. Following MOMP, which is considered as a point of no return, mitochondrial proteins such as cytochrome c (cyt c) are released into the cytosol, where they activate caspase proteases and programmed cell death.

In addition to its widely recognized apoptotic role, MOMP has also been linked to important pro-inflammatory responses [119]. The disruption of MOM integrity can lead to cytosolic exposure of mtROS and danger-associated molecular patterns (DAMPs) such as mtDNA. Once exposed to the cytosol, mtDNA can trigger the activation of endosomal localized toll-like receptor 9 (TLR9) and cyclic GMP-AMP synthase (cGAS) stimulator of interferon genes (STING), resulting in type I interferon responses [120]. Furthermore, oxidized mtDNA is able to induce nucleotide-binding domain leucine-rich repeat family, pyrin domain containing 3 (NLRP3) inflammasome formation and consequent IL-1β and IL-18 secretion, thereby activating neutrophils, macrophages, and T cells.

It has been reported that apoptotic caspases have the capacity to inhibit mitochondria-dependent inflammatory effects during cell death [121]. In contrast, by downregulating the inhibitor of apoptosis proteins, mitochondrial proteins can initiate NF-κB signaling, resulting in the activation of transcription of several inflammatory genes [122]. This means that mitochondria define the choice and either induce an apoptotic immunosilent or an inflammatory type of cell death.

Thus, as major regulators of apoptosis and inflammation, mitochondria could play an important role in the FLASH effect, in which apoptotic and inflammatory responses have been shown to be significantly reduced compared to CONV-dose-rate irradiation. If FLASH RT could reduce mitochondrial damage and/or mtROS levels, greater normal tissue sparing might be achieved (Figure 2). Decreased induction of mtROS would only cause small changes in ROS homeostasis, leaving normal cells viable and functional. However, malignant cells are characterized by mtROS overproduction, which promotes metabolic reprogramming, genomic instability, and tumorigenic signaling, making them potentially more susceptible to changes in ROS levels [123,124]. Even small increases in mtROS might result in a critical excess of ROS in cancer cells and, as a result, increased cancer cell death. This may possibly explain why tumor cells do not benefit from the FLASH effect and a similar tumor control is observed after FLASH vs. CONV RT.

Recently, the effect of proton FLASH irradiation on mitochondrial function was investigated in normal human lung fibroblasts (IMR90) that were exposed to 15 Gy at either FLASH (100 Gy/s) or CONV (0.33 Gy/s) dose rates under an ambient oxygen concentration (21%) [125]. Compared to CONV irradiation, FLASH RT preserved survival and induced minimal mitochondrial damage characterized by morphological changes, functional changes (membrane potential, mtDNA copy number, and cellular ATP levels), and ROS production. In contrast, cell viability and mitochondrial morphology of lung cancer cells (A549) were negatively affected by both FLASH- and CONV-dose-rate irradiation. These data indicated that FLASH RT could spare normal cells from mitochondrial damage but not cancer cells. Another in vitro study using mouse embryonic fibroblast cells revealed that late apoptosis and necrosis were significantly decreased in FLASH (>10^9^ Gy/s ultra-fast laser-generated particles)-irradiated cyt c^−/−^ cells compared to cyt c^+/+^ cells in both normoxic conditions and hypoxia-like conditions induced by CoCl_2_ addition [126]. The difference in late apoptosis and necrosis between cyt c^−/−^ and cyt c^+/+^ cells was also more pronounced after FLASH RT (29% decrease on average) relative to CONV (0.05 Gy/s Co_60_ γ-radiation) treatment (14.25% decrease on average). This means that a reduction in cyt c release, most likely as a consequence of less mitochondrial damage, could at least partially be responsible for the observed FLASH effect. To validate these results, cyt c release from mitochondria into the cytosol should be directly measured following exposure to FLASH and CONV dose rates.

In the in vivo setting, the reduced mitochondrial damage might also result from the potential sparing of circulating immune cells by ultra-high dose rate irradiation. Additionally, if FLASH RT could preserve the function of T cells and their mitochondria, its combination with immunotherapy, which often depends on functioning T cells, could be a promising strategy to enhance anti-tumor immunity [127].

As mitochondrial damage and its consequences have not been studied thoroughly in the context of the FLASH effects so far, comparable studies on mitochondrial response, especially mtDNA and mtROS release, after FLASH RT in normal and cancer cells will be necessary to provide substantial evidence.

## 3. Beam Parameters Necessary to Trigger the FLASH Effect

In addition to a good understanding of the biological mechanisms, the characterization of the exact beam parameters necessary to trigger the FLASH effect is required for optimal translation of ultra-high dose rate RT into the clinic. An average dose rate above 40 Gy/s has long been the main characteristic that defines FLASH RT. However, it seems that a higher dose rate is often needed to observe a reduction in normal tissue toxicity, depending on the target tissue (brain [24,32,34] and abdomen [25,42,44] vs. thorax [17,36,37,38]) and the used in vivo model (zebrafish embryos [22] vs. mice [31,46]). In addition to the dose rate, a dose threshold also strongly impacts the FLASH effect. While the majority of preclinical FLASH studies have administered the radiation dose in large single fractions (>10 Gy), clinical implementation of such doses would currently be unattainable. Interestingly, a recent study demonstrated that hypofractionated FLASH RT (1.8 × 10^6^ Gy/s; 3 × 10 Gy) was as efficient as CONV RT (0.1 Gy/s; 3 × 10 Gy) in delaying glioblastoma growth in mice and spared the normal brain from radiation-induced toxicities [27]. The radiation modality should also be considered when evaluating the FLASH effect.

Most preclinical studies on FLASH have used electron beams; however, proton, X-ray, and carbon ion ultra-high dose rate irradiation are also showing promising results. Hadron therapy (protons and carbon ions) is characterized by unique ballistic properties that allow for a lower dose deposition in healthy tissues and a maximum dose deposition in the tumor. In addition to that, carbon ions (densely ionizing or high-LET particles) are characterized by a higher relative biological effectiveness, resulting in increased cell killing. The radiobiological effect of high-LET carbon ions depends much less on the oxygen enhancement ratio compared to sparsely ionizing radiation [128]. Moreover, carbon ions generate an oxygenated microenvironment around their track, especially in the Bragg peak region [129]. Even though the exact mechanism remains to be elucidated, it is clear that the oxygenation level does play a role in the FLASH effect. Thus, experimental preclinical carbon ion studies are of pivotal importance not only for the potential implementation of the carbon FLASH therapy in the clinical practice, but also for the understanding of the FLASH effect. The experimental results obtained so far were in good accordance with the FLASH studies in which sparsely ionizing radiation was used. The FLASH effect was not observed for carbon ion irradiation in vitro under ambient oxygen concentrations in HFL1 lung fibroblasts and HSGc-C5 salivary gland cancer cells [59] or in CHO-K1 cells under anoxic and normoxic conditions. However, the protective effect was observed in the CHO-K1 cells at 0.5% and 4% O_2_ concentrations [58]. The first in vivo FLASH carbon ion study showed reduced normal tissue toxicity, comparable tumor control, and a superior reduction in lung metastases in an osteosarcoma mouse model [48]. Even though all the studies performed so far irradiated in the plateau region before the Bragg peak, the LET values (13–50 keV/µm) were still significantly higher than that of protons, electrons, or X-rays. Thus, it can be concluded that the FLASH effect can also be triggered by densely ionizing radiation.

For proton irradiations, the beam pulse structure might also play a determining factor in the observation of the FLASH effect. The recent findings of Karsch et al. indicated that FLASH quasi-continuous beam delivery could result in lower normal tissue toxicity of zebrafish embryos compared to the FLASH macro pulse delivery [49]. Furthermore, the dose rate within the pulse (≥10^6^ Gy/s), dose-per-pulse (≥1 Gy), and overall treatment time (≤0.1s) may also influence the FLASH effect [52].

## 4. Conclusions

Ultra-high dose rate “FLASH” radiotherapy has shown a great potential to improve the therapeutic index for cancer treatment. Its unique normal-tissue-sparing effects and comparable tumor control relative to CONV RT have already been confirmed in skin, lung, brain, and intestine in several in vivo models and even in a first patient with T-cell cutaneous lymphoma. Moreover, novel innovations in RT technology have enabled researchers to conduct FLASH clinical trials in an attempt to translate those preclinical findings to the clinic. The first-in-human clinical trial of FLASH therapy, FAST-01, has begun at the University of Cincinnati to evaluate the feasibility and safety of single-fraction proton FLASH RT for painful bone metastases [130]. A second FLASH clinical trial at the Lausanne University Hospital involves electron-dose-escalation to determine the FLASH dose that will control skin metastases from melanoma, a radioresistant type of cancer [131].

Even though FLASH RT may become one of the greatest breakthroughs in the radiation oncology field, the mechanisms behind the FLASH effect are still poorly understood. One of the first hypotheses stated that an acute depletion of oxygen following ultra-high dose rate irradiation protects healthy tissues from RT-induced DNA damage, while FLASH RT has less of an impact on the radiosensitivity in tumors, as they are already hypoxic. However, recent observations have suggested that a complete oxygen depletion by FLASH RT is unlikely to occur in well-oxygenated normal tissues and therefore cannot be the sole explanation for the FLASH effect [72,132,133]. Additionally, a modified immune response has also been introduced to explain the differential impact of FLASH vs. CONV RT on circulating immune cells, TME, cytokine signaling, and inflammatory processes. The immunomodulatory effects of FLASH RT may not only explain the reported reduction in inflammation and fibrosis, but also the maintenance of a similar tumor control relative to CONV RT through the induction of a more potent anti-tumor immunity. However, more thorough investigations are needed to clarify the differential impact of ultra-high dose rates on the immune factors in tumors and their surrounding normal tissues in comparison with standard treatment. Finally, a mitochondria-mediated mechanism might be the missing link between FLASH dose rates and the observed normal tissue sparing in preclinical studies. Significantly more research is required to characterize the potential role of mitochondria and other factors in the FLASH effect in order to optimize the physical parameters to set more effective and less toxic clinical RT approaches.

We believe that many cancer patients could potentially benefit from FLASH radiotherapy in the future, especially those with radioresistant tumors in need of dose escalation, patients with tumors closely located to organs at risk, and patients with higher normal tissue radiosensitivity. However, it will take several years before FLASH RT can be successfully implemented in the clinic because several challenges are still to be overcome: the development of clinical devices capable of delivering doses at ultra-high dose rates to large tissue volumes, advances in dosimetry technology, validation of (hypo)fractionated FLASH therapy, etc. In addition, clinical trials are necessary to monitor both acute and late toxicity effects in different organs and to assess the quality and safety of this new treatment approach. These clinical data will also improve our understanding of FLASH radiobiology and bring us one step closer to optimal exploitation of its radiotherapy potential.

## Figures and Tables

**Figure 1 ijms-23-12109-f001:**
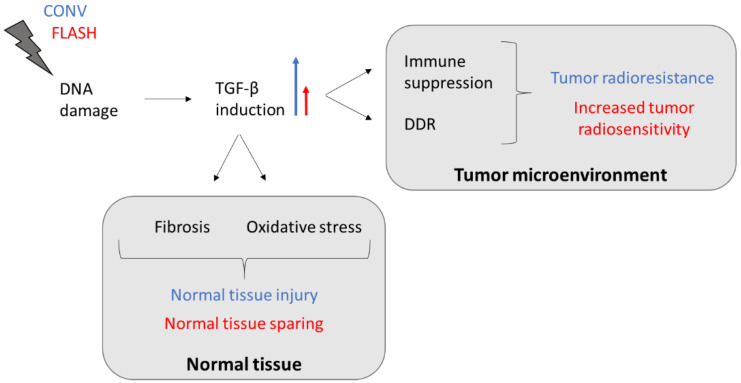
TGF-β-induced tumor resistance and normal tissue injury following CONV RT might be reduced by FLASH-dose-rate irradiation. The blue long arrow indicates a great induction (CONV) and the red short arrow indicates a decreased induction (FLASH). Abbreviations: TGF-β, transforming growth factor beta; DDR, DNA damage response.

**Figure 2 ijms-23-12109-f002:**
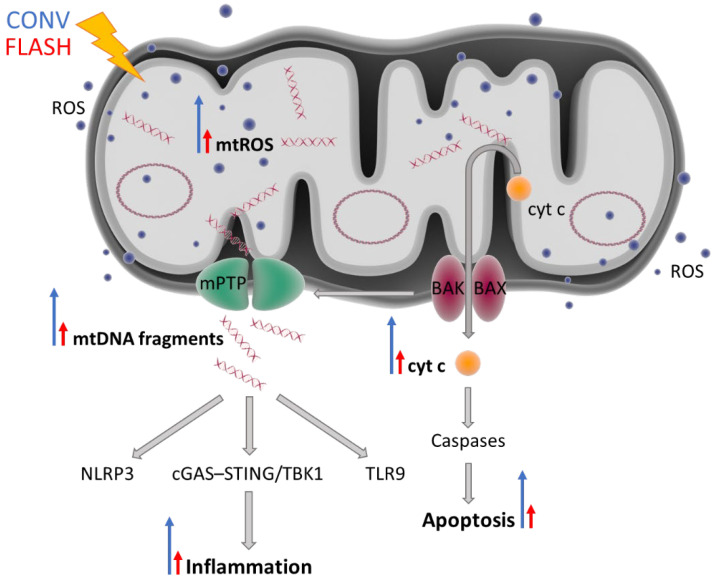
FLASH irradiation might induce minimal mitochondrial damage, resulting in less inflammation and less apoptosis compared to CONV RT. Radiation can damage mtDNA either by directly interacting with it or indirectly via ROS generated as the result of water radiolysis. Dysfunctional mitochondria produce increased levels of endogenous ROS. Mitochondrial dysfunction and overall cellular stress result in activation of pro-apoptotic BAK and BAX proteins, which form pores in the mitochondrial outer membrane and promote the release of cytochrome c into the cytoplasm, thus initiating a cascade of apoptotic events. BAK and BAX also induce mitochondrial outer membrane permeabilization (MOMP) and are required for the opening of the mitochondrial permeability transition pore. MOMP enables the release of damage-associated molecular patterns such as fragmented or oxidized mtDNA, which can trigger inflammation via the activation of three main pro-inflammatory pathways: NLRP3, cGAS–STING/TBK1, and TLR9. The blue long arrows indicate a great induction (CONV) and the red short arrows indicate a decreased induction (FLASH). The blue dots and oval circles represent ROS and circular mtDNA, respectively. Abbreviations: BAK, Bcl2 homologous antagonist/killer; BAX, BCL-associated X; cGAS–STING, GMP/AMP synthase–stimulator of interferon genes DNA-sensing system; cyt c, cytochrome c; mPTP, mitochondrial permeability transition pore; mtDNA, mitochondrial DNA; NLRP3, nucleotide-binding oligomerization domain-like receptor family pyrin domain containing 3; ROS, reactive oxygen species; TLR9, toll-like receptor 9; TBK1, TRAF family-member-associated NF-κB activator-binding kinase 1.

**Table 1 ijms-23-12109-t001:** Preclinical evidence of normal tissue sparing and equivalent tumor control after FLASH RT.

Model (Site of Irradiation)	FLASH RT (Gy/s)	CONV RT (Gy/s)	Dose (Gy)	Radiation Modality	Main Findings		Ref. Year
					Normal Tissue	Tumor	
Mice (WBI ^1^)	>100	0.1	10	Electron	Preserved spatial memory and neurogenesis in hippocampus	-	[34] 2017
Mice (WBI)	>100	0.07–0.1	10	Electron	Preserved cognitive function, neuronal morphology, and dendritic spine density; no neuroinflammation	-	[21] 2019
Mice (WBI)	200–300	0.13	30	Electron	Less hippocampal dendritic spine loss and neuroinflammation	-	[32] 2019
Mice (WBI)	>4.4 × 10^6^	0.1	8	Electron	Preserved developing/mature neurons; minimized microgliosis; limited reduction of plasmatic growth hormone levels	-	[30] 2020
Mice (WBI)	>106	0.09	10–25	Electron	Reduced levels of apoptosis; reduced vascular dilation; preserved microvasculature integrity	-	[24] 2020
Mice (WBI)	1.8 × 10^6^	0.1	3 × 10	Electron	Sparing of cognitive deficits in learning and memory	Similar tumor control	[27] 2020
Mice (WBI)	5.6 × 10^6^	0.1	10	Electron	Reduced astrogliosis and immune signaling in the brain	-	[28] 2020
Mice (WBI)	37	0.05	10	Photon	No memory deficit; less hypocampal cell division- impairment; less reactive astrogliosis	-	[33] 2018
Mice (thorax)	≥40	≤0.03	17	Electron	No lung fibrosis; sparing of normal smooth muscle and epithelial cells	Isoefficient tumor inhibition	[17] 2014
Mice (thorax)	40–60	≤0.03	17	Electron	Minimized induction of pro-inflammatory genes; less persistent DNA damage and senescent cells; sparing of lung progenitor cells from excessive damage	-	[31] 2020
Mice (thorax)	180	0.07	30–40	Electron	Reduced severe skin toxicity; reduced mortality	-	[29] 2020
Mice (thorax)	352	0.06	15	Electron	-	No constricted vessel morphology; reduced p-MLC expression; reduced yH2AX-positive cells and more ROS in tumors	[40] 2021
Mice (thorax)	700	0.1	30	Photon	Improved survival	-	[39] 2021
Mice (thorax)	40	1	15–17.5–20	Proton	Reduced lung fibrosis and skin dermatitis; improved mouse survival	-	[38] 2019
Mice (thorax)	40	0.5	17.5–20	Proton	Lower incidence of dermatitis; better breathing function; better overall survival	-	[36] 2019
Mice (thorax)	40	0.5	18	Proton	-	Smaller lung tumors; improved recruitment of CD3+ T cells into tumor	[37] 2019
Mice (abdomen)	70–210	0.05	10–22	Electron	Improved survival	-	[41] 2017
Mice (abdomen)	216	0.079	12–16	Electron	Reduced radiation-induced intestinal injury; spared gut function and epithelial integrity; less cell death in crypt base columnar cells	Similar efficacy in reducing tumor burden	[25] 2020
Mice (abdomen)	≥280	0.25	7.5–12.5	Electron	Improved crypt survival; fewer microbiota changes	-	[42] 2021
Mice (abdomen)	210	0.126	14	Electron	Enhanced intestinal regeneration; reduced T-reg cells; increased cytolytic T cells	Isoefficient tumor control	[43] 2022
Mice (abdomen)	700	0.1	12	Photon	Improved survival	-	[39] 2021
Mice (abdomen)	>150	0.1	10–15	Photon	Faster body weight recovery; higher survival probability; less acute intestinal damage; fewer inflammatory blood cells and diminished pro-inflammatory cytokines; reduced lipid peroxidation	-	[44] 2022
Mice (abdomen)	78	0.9	15–18	Proton	Reduced loss in proliferating cells in intestinal crypts; less intestinal fibrosis	Isoefficient tumor inhibition	[16] 2020
Mice (abdomen)	>107	0.82	15–18	Proton	More EdU+/crypt cells and regenerating crypts; improved survival	Isoefficient tumor inhibition	[23] 2021
Mice (limb)	69–124	0.39–0.65	30	Proton	Reduced skin injury, stem cell depletion, and inflammation; mitigated lymphedema; decreased myofiber atrophy, bone resorption, hair follicle atrophy, and epidermal hyperplasia; less TGFβ1	Equipotent in sarcoma control	[45] 2021
Mice (limb)	80	0.37	23–50	Proton	Less skin toxicity	-	[46] 2022
Mice (limb)	83	0.38	40–60	Proton	Reduced acute skin damage and radiaiton-induced fibrosis	Similar tumor control	[47] 2022
Mice (limb)	100	0.3	18	Carbon	Reduced structural changes in muscle	Similar tumor control; fewer lung metastases	[48] 2020
Mice (breast)	1000	0.1	18 (FLASH) 15 (CONV)	Photon	-	Slower increase in tumor volume	[39] 2021
Mice (total body)	200	<0.072	4	Electron	Reduced funtional damage to human blood cells	Similar tumor control in 2 T-ALL cases	[26] 2021
Mini-pigs (skin)	300	0.083	22–34	Electron	Reduced skin toxicity/injury	-	[15] 2019
Zebrafish embryos	>100	>0.1	8	Electron	Fewer alterations in body length	-	[21] 2019
Zebrafish embryos	10^5^	0.1	26	Electron	Longer bodies; fewer embryos with spinal curvature and pericardial edema	-	[22] 2021
Zebrafish embryos	300	0.15	30	Proton	Longer bodies; fewer embryos with spinal curvature and pericardial edema	-	[49] 2022

^1^ WBI: whole-brain irradiation.

**Table 2 ijms-23-12109-t002:** Summary of in vitro studies showing a FLASH effect.

Cell Line	FLASH RT (Gy/s)	CONV RT (Gy/s)	Dose (Gy)	Radiation Modality	Results	Ref. Year
Prostate cancer cells (DU145)	600	0.23	18	Electron	Increased survival in hypoxic conditions (1.6–4.4% O_2_)	[54] 2019
Human lung fibroblasts (MRC50; IMR90)	≥40	≤0.03	5	Electron	Reduced DNA damage induction and lethality	[31] 2020
Human pulmonary basalepithelial cells (PBEC)	≥40	≤0.03	4	Electron	Sparing from radiation-induced differentiation and cell death	[31] 2020
Human breast cancer cells (MDA-MB-231; MCF7); human cervix cancer cells (HeLa)	800	0.23	6–10	Electron	Increased survival in normoxic conditions	[55] 2021
Lung adenocarcinoma spheroids (A549)	90	0.075	10	Electron	3-fold higher clonogenic survival	[56] 2021
Normal human lung fibroblasts (IMR90)	1000	0.05–0.2	20	Proton	Less yH2AX foci formation; fewer senescence cells; less TGFβ1 induction	[57] 2019
Chinese hamster ovary cells (CHO-K1)	70	0.6	7.5	Carbon	Increased survival in hypoxic conditions (0.5–4%)	[58] 2022
Normal human lung fibroblasts (HFL1); human salivary gland cancer cells (HSGc-C5)	96–195	8–13	1, 2, 3	Carbon	No difference in growth suppression and senescence of HFL1 cells and in survival of HSGc-C5 cells in normoxic conditions	[59]

## Data Availability

Not applicable.

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
