# Peer review of "Potential Molecular Mechanisms behind the Ultra-High Dose Rate “FLASH” Effect"

_ijms, 2022, doi:10.3390/ijms232012109_

Round 1
Reviewer 1 Report
This paper describes the molecular mechanism of FLASH and is very interesting.
The proton beam is currently the only clinically viable device for FLASH. However, in the future, electron beams may lead to the development of clinically viable devices.
Proton beams and electron beams have different treatment mechanisms, and proton beam FLASH and electron beam FLASH should also have different mechanisms.
Since the paper cites FLASH literature on proton beams, electron beams, and carbon beams, it is desirable to add a discussion on how differences in beam types affect the mechanism.
Reviewer 2 Report
the paper is a review of the biological effects behind the FLASH method which is a relatively new concept in the radiotherapy field. The authors provide a good overview of the mechanisms involved in FlASH and other studies done in this field. I recommend this paper for publication in the present form at the IJMS.
some comments for improving the paper:
1. in Table 1: the outcome of study 37 is not mentioned. please complete this part in the table.
2. adding some figures for section 2.2 and other sections helps clarify the concept to the readers.
3. in the conclusion, please add your point of view on FLASH in cancer treatment
